# ADIABATIC REPLAY FOR CONTINUAL LEARNING

## ABSTRACT

To avoid catastrophic forgetting, many replay-based approaches to continual learning (CL) require, for each learning phase with new data, the replay of samples representing *all* of the previously learned knowledge. Since this knowledge grows over time, such approaches invest linearly growing computational resources just for re-learning what is already known. In this proof-of-concept study, we propose a generative replay-based CL strategy that we term adiabatic replay (AR), which achieves CL in constant time and memory complexity by making use of the (very common) situation where each new learning phase is *adiabatic*, i.e., represents only a small addition to existing knowledge. The employed Gaussian Mixture Models (GMMs) are capable of *selective updating* only those parts of their internal representation affected by the new task. The information that would otherwise be overwritten by such updates is protected by *selective replay* of samples that are similar to newly arriving ones. Thus, the amount of to-be-replayed samples depends not at all on accumulated, but only on added knowledge, which is small by construction. Based on the challenging CIFAR, SVHN and Fruits datasets in combination with pre-trained feature extractors, we confirm AR's superior scaling behavior while showing better accuracy than common baselines in the field.

## 1 INTRODUCTION

This contribution is in the context of continual learning (CL), a recent flavor of machine learning that investigates learning from data with non-stationary distributions. A common effect in this context is catastrophic forgetting (CF), an effect where previously acquired knowledge is abruptly lost after a change in data distributions. In class-incremental CL (see, e.g., Bagus et al. (2022); van de Ven et al. (2022)), a number of assumptions are made: distribution changes are assumed to be abrupt, partitioning the data stream into stationary *tasks*. Then, task onsets are supposed to be known, instead of inferring them from data. Lastly, tasks are assumed to be disjoint. Together with this goes the constraint that no, or only a few, samples may be stored. A very promising approach to

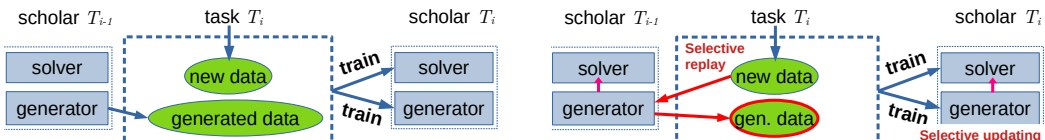

Figure 1: Left: schematics of generative replay. A *scholar* composed of generator and solver is trained at every task. The solver performs the task, e.g., classification, whereas the generator serves as a memory for samples from previous tasks $T_{i'}$, $i' < i$. Please note that the amount of generated data usually far exceeds the amount of new data. Right: schematics of adiabatic replay, red indicates differences to generative replay. At every task, new data is used to query the generator, therefore generated data are produced in constant proportion. Furthermore, the generator additionally serves a feature generator for the solver, thus saving computational resources.

mitigate catastrophic forgetting (CF) in this scenario are replay strategies van de Ven et al. (2020). *Replay* aims at preventing CF by using samples from previous tasks to augment the current one. On the one hand, there are "true" replay methods which use a small number of stored samples for augmentation. On the other hand, there are *pseudo-replay* methods, where the samples to augment the current task are produced in unlimited number by a generator, which removes the need to store

samples. A schematics of the training process in generative replay is given in Fig. 1. Replay, in its original formulation, proposes a principled approach to CL, but it nevertheless presents several challenges: First of all, if DNNs are employed as solvers and generators, then all classes must be represented in equal proportion at every task in order to have any kind of performance guarantees. Thus, for example, in the simple case of a single new class ($D$ samples) per task, the generator must produce $(s-1)D$ samples at task $T_s$ in order to always train with $D$ samples per class. This unbounded linear growth of to-be-replayed samples, and therefore of training time, as a function of the number of previous tasks $s$, poses enormous problems for long-term CL. For instance, even very small additions to a large body of existing knowledge (a common use-case) require a large amount of samples to be replayed, see, e.g., Dzemidovich & Gepperth (2022). Since replay is always a lossy process, this imposes severe limits on GR-based CL performance.

## 1.1 Approach: AR

Adiabatic replay (AR) prevents an ever-growing number of replayed samples by applying two main strategies: selective replay and selective updating, see Fig. 1. Selective replay means that new data are used to query the generator for similar (potentially conflicting) samples. For achieving selective replay, we rely on Gaussian Mixture Models (GMMs) trained by SGD as introduced in Gepperth & Pfülb (2021a). GMMs have limited modeling capacity, but are sufficient here since we are working with pre-trained feature extractors.

AR is partly inspired by maximally interfered retrieval (MIR), proposed in Aljundi et al. (2019b) where a fixed replay budget (either for experience replay or generative replay) is composed of the most conflicted samples, those that would be unlearned most rapidly when training on a new task. In a similar vein, McClelland et al. (2020) hypothesize that just replaying samples that are similar to new ones could be sufficient to avoid forgetting. Another inspiration comes from Klasson et al. (2023), where it is shown that replaying the right data at the right moment is preferable to replaying everything.

Adiabatic replay is most efficient when each task $t$, with data $\vec{x} \sim p^{(t)}$, adds only a small amount of new knowledge. AR models the joint distribution of past tasks as a mixture model with $K$ components, $p^{(1...t)}(\vec{x}) = \sum_k \pi_k \mathcal{N}(\vec{x}; \vec{\mu}_k, \mathbf{\Sigma}_k)$, thus we can formalize this assumption as a requirement that only a few components in $p^{(1...t)}$ are activated by new data: $|\{\arg\max_k \pi_k \mathcal{N}(\vec{x}_i; \vec{\mu}_k, \mathbf{\Sigma}_k) \forall \vec{x}_i \sim p^{(t)}\}| \ll K$. If this assumption is violated, AR will still work but more components will need to be updated, requiring more samples. It was demonstrated in Gepperth & Pfülb (2021a) that GMMs have an intrinsic capacity for *selective updating* when re-trained with new data. Concretely, only the components that are similar to, and thus potentially in conflict with, incoming data are adapted. In contrast, dissimilar components are not adapted, and are thus protected against CF.

## 1.2 Contributions

**Selective replay:** Previous knowledge is not replayed indiscriminately, but only where significant overlap with new data exists.

**Selective updating:** Previous knowledge is only modified by new data where an overlap exists.

**Near-Constant time complexity:** Assuming that each task adds only a small fraction to accumulated knowledge (adiabatic assumption), the number of generated/replayed samples can be small as well, and in particular does not grow with the number of tasks.

**Integration of pre-trained feature extractors:** To process visual problems of higher complexity (SVHN, CIFAR), we incorporate recent advances in latent replay into AR, where we do not replay raw samples but higher-level representations generated by a frozen feature extractor network.

## 1.3 Related Work

In recent years, diverging strategies were presented to mitigate CF in CL scenarios, please refer to De Lange et al. (2021); Masana et al. (2022); Hadsell et al. (2020); Lesort et al. (2020) for an overview. Broad strategies include regularization, parameter isolation and rehearsal. In this article, we focus on rehearsal-type CL, and in particular on deep generative replay (DGR).

**Rehearsal/Replay** This branch of CL solutions relies on the storage of previously encountered data instances. In its purest form, past data is held inside a buffer and mixed with data from the current task to avoid CF, as shown in Gepperth & Karaoguz (2016); Rebuffi et al. (2017); Rolnick et al. (2019); De Lange & Tuytelaars (2021). This has drawbacks in practice, since it breaks the constraints for task-incremental learning Van de Ven & Tolias (2019), has privacy concerns, and requires significant memory. *Partial replay*, e.g. Aljundi et al. (2019a), and constraint-based optimization Lopez-Paz & Ranzato (2017); Chaudhry et al. (2018; 2019); Aljundi et al. (2019c), focus on selecting a subset from previously seen samples, but it appears that the selection of a sufficient subset is still challenging Prabhu et al. (2020). Comprehensive overviews about current advances in replay can be found in Hayes et al. (2021); Bagus & Gepperth (2021).

**Deep Generative Replay** Here, (deep) generative models like GANs Goodfellow et al. (2014) and VAEs Kingma & Welling (2013) are used for memory consolidation by replaying samples from previous tasks, see Fig. 1 and Shin et al. (2017). The recent growing interest in GR brought up a variety of architectures, either being VAE-based Kamra et al. (2017); Lavda et al. (2018); Ramapuram et al. (2020); Ye & Bors (2020); Caselles-Dupré et al. (2021) or GAN-based Ostapenko et al. (2019); Wang et al. (2021); Atkinson et al. (2021). Notably, the MerGAN model Wu et al. (2018) uses an LwF-type knowledge distillation technique to prevent forgetting in generators, which is more efficient than pure replay. Furthermore, PASS Zhu et al. (2021) uses self-supervised learning by sample augmentation in conjunction with slim class-based prototype storage for improving the performance replay-based CL. An increasingly employed technique in this respect is *latent replay* which operates on and replays latent features generated by a frozen encoder network, see, e.g., van de Ven et al. (2020); Pellegrini et al. (2020); Kong et al. (2023). Built on this idea are models like RE-MIND Hayes et al. (2020), which extends latent replay by the aspect of compression, or SIESTA Harun et al. (2023) which improves computational efficiency by alternating wake and sleep phases in which different parts of the architecture are adapted.

**MIR** Conceptually, this is similar to the concept of selective replay, although a key difference is that our GMM generator/solver is capable of selective updating as well. We will use MIR as one of the baselines for our experiments.

## 2 METHODS

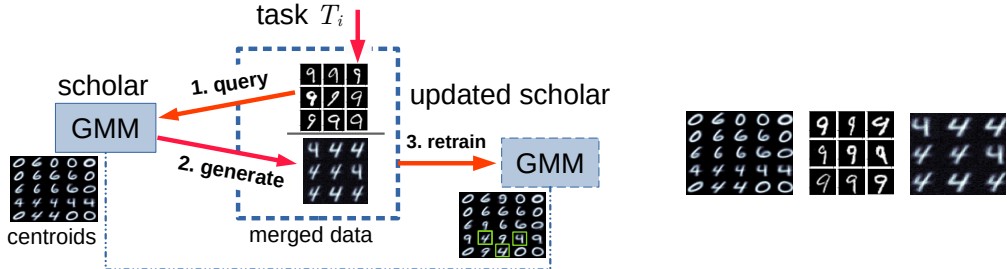

Figure 2: Left: The proposed AR approach, illustrated in an exemplary MNIST setting. The scholar (GMM) has been trained on MNIST classes 0, 4 and 6 in task $T_1$. At task $T_2$, new data (class 9) is used to *query* the scholar for similar samples, resulting in the selective replay of mostly 4's but no 0's. The scholar is re-trained *from its current state*, so no data concerning class 0 is required. Re-training results in the insertion of 9's into the existing components. This mechanism works identically for higher-level features produced by a pre-trained feature extractor. Right: enlarged GMM prototypes, query samples and variant generation results.

The main techniques used in the experiments of this article are adiabatic replay (AR), experience replay (ER), deep generative replay (DGR) and pre-trained feature extractors. We refer to the appendix for details on the experimental settings concerning ER (Appendix D), DGR (Appendix C) and the encoding of data by pre-trained models (Appendix A), whereas we will discuss the details of AR in this section.

## 2.1 ADIABATIC REPLAY (AR)

In contrast to conventional replay, where a scholar is composed of a generator and a solver network, see Fig. 1, AR proposes scholars where a single network acts as a generator as well as a feature generator for the solver. Assuming a suitable scholar (see below), the high-level logic of AR is shown in Fig. 2: Each sample from a new task is used to *query* the scholar, which generates a similar, known sample. Mixing new and generated samples in a defined, constant proportion creates the training data for the current task (see Algorithm 1 for pseudocode). A new sample will cause adaptation of the scholar in a localized region of data space. Variants generated by that sample will, due to similarity, cause adaptation in the same region. Knowledge in the overlap region will therefore be adapted to represent both, while dissimilar regions stay unaffected (see Fig. 2 for a visual impression).

None of these requirements are fulfilled by DNNs, which is why we implement the scholar by a "flat" GMM layer (generator/feature encoder) followed by a linear classifier (solver). Both are independently trained via SGD according to Gepperth & Pfülb (2021a). Extensions to deep convolutional GMMs (DCGMMs) Gepperth (2022) for higher sampling capacity can be incorporated as drop-in replacements for the generator.

**Data:** AR scholar/gen. $\Phi$, AR solver $\Theta$, real data $\mathcal{X}^t$, $Y^t$
**for** $t \in 2...T$ **do**
   **for** $\mathcal{B}_N \sim \mathcal{X}_t$ **do**
      // Propagate batch $\mathcal{B}_N$ though $\Phi$.
      $\sigma_{\mathcal{B}_N} \leftarrow \Phi(\mathcal{B}_N)$;
      // Query batch of variants from $\Phi$.
      $\mathcal{B}_G \leftarrow Vargen(\Phi, \sigma_{\mathcal{B}_N})$;
      // Add gen. samples to $\mathcal{X}_G^t$.
      $\mathcal{X}_G^t \leftarrow UpdateData(\mathcal{B}_G)$
   **end**
   **for** $\mathcal{B}_M \sim (\mathcal{X}^t \cup \mathcal{X}_G^t)$ **do**
      // Update $\Phi$ and $\Theta$
      $\Phi \leftarrow SGD(\mathcal{B}_M)$;
      $\Theta \leftarrow SGD(\Phi(\mathcal{B}_M), Y_t)$;
   **end**
**end**

**Algorithm 1:** Adiabatic Replay

**Selective updating** is an intrinsic property of GMMs. They describe data distributions by a set of $K$ *components*, consisting of component weights $\pi_k$, centroids $\boldsymbol{\mu}_k$ and covariance matrices $\boldsymbol{\Sigma}_k$. A data sample $\boldsymbol{x}$ is assigned a probability $p(\boldsymbol{x}) = \sum_k \pi_k \mathcal{N}(\boldsymbol{x}; \boldsymbol{\mu}_k, \boldsymbol{\Sigma}_k)$ as a weighted sum of normal distributions $\mathcal{N}(\boldsymbol{x}; \boldsymbol{\mu}_k, \boldsymbol{\Sigma}_k)$. Training of GMMs is performed as detailed in Gepperth & Pfülb (2021a) by adapting centroids, covariance matrices and component weights through the SGD-based minimization of the negative log-likelihood $\mathcal{L} = \sum_n \log \sum_k \pi_k \mathcal{N}(\boldsymbol{x}_n; \boldsymbol{\mu}_k, \boldsymbol{\Sigma}_k)$. As shown in Gepperth & Pfülb (2021a), this expression is strongly dominated by a single GMM component $k^*$, and can be approximated as $-\log(\pi_{k^*} \mathcal{N}(\boldsymbol{x}; \boldsymbol{\mu}_{k^*}, \boldsymbol{\Sigma}_{k^*}))$. This implies that the best-matching GMM component $k^*$ is the only component that selectively adapted.

**Selective replay** is a form of sampling from the probability density represented by a trained GMM, see Gepperth & Pfülb (2021b). It is triggered by a query in the form of a data sample $\boldsymbol{x}_n$, which is converted into a control signal $\mathcal{T}$ defined by the posterior probabilities (or *responsibilities*):

$$\gamma_k(\boldsymbol{x}_n) = \frac{\pi_k \mathcal{N}(\boldsymbol{x}_n; \boldsymbol{\mu}_k, \boldsymbol{\Sigma}_k)}{\sum_j \pi_j \mathcal{N}(\boldsymbol{x}_n; \boldsymbol{\mu}_j, \boldsymbol{\Sigma}_j)}. \tag{1}$$

For selective replay, these responsibilities parameterize a multinomial distribution for drawing a GMM component $k^*$ to sample from, instead of the component weights $\pi_K$ as usually done in GMM sampling. To reduce noise, top-S sampling is introduced, where only the $S = 3$ highest values of the responsibilities are used for selection.

**Solver** functions are performed by feeding GMM responsibilities into a linear regression layer as $o(x_n) = W\gamma(x_n)$. We use a MSE loss and drop the bias term to reduce the sensitivity to unbalanced classes.

**GMM training** uses the procedure described in Gepperth & Pfülb (2021a) including the recommended values for learning rate and regularization. Details of the training procedure and the general AR architecture are given in Appendix B.

## 3 EXPERIMENTS

### 3.1 EVALUATION DATA AND FEATURE ENCODING

**MNIST** LeCun et al. (1998) consists of $60.000$ $28 \times 28$ grayscale images of handwritten digits (0-9).

**Fashion-MNIST** Xiao et al. (2017) consists of 60.000 images of clothes in 10 categories and is structured like MNIST.

**E-MNIST** Cohen et al. (2017) is structured like MNIST and extends it by letters. We use the balanced split which contains 131.000 samples in 47 classes.

**Fruits-360** Mureșan & Oltean (2018) contains 100x100 images showing different types of fruits, from which we chose the 10 best-represented classes and downsample to 32x32 RGB.

**SVHN** Netzer et al. (2011) contains 60.000 RGB images of house numbers (0-9, resolution $32 \times 32$).

**CIFAR-10** Krizhevsky et al. (2009) contains 60.000 RGB images of natural objects, resolution 32x32, in 10 balanced classes.

We construct the following CIL problems by splitting the datasets as follows: D5-1$^5$A (6 tasks, 0-4,5,6,7,8,9), D5-1$^5$B (6 tasks, 5-9,0,1,2,3,4), D7-1$^3$A (4 tasks, 0-6,7,8,9), D7-1$^5$B (4 tasks, 3-9,0,1,2), D20-1$^5$A (6 tasks, 0-19,20,21,22,23,24, EMNIST only) and D2-2$^5$ (5 tasks, 0-1,2-3,4-5,6-7,8-9).

D20-1$^5$A for EMNIST represents a CL problem where the amount of already acquired knowledge is significantly larger than the amount of new data added with each successive task.

No feature encoding is performed for MNIST, Fashion-MNIST, E-MNIST and Fruits-360 due to their inherent simplicity. The encoding of SVHN and CIFAR is described in Appendix A.

### 3.2 EVALUATION MEASURES

Similar to Kemker et al. (2018); Mundt et al. (2021), we provide the final (averaged) accuracy $\alpha_T$, evaluating a scholar $\mathcal{S}_T$ on a test set $T_{\text{ALL}}$ after full training on each sub task $T$ for any given class-incremental learning problem (CIL-P) listed in Sec. 3.1. The values are normalized to a range of $\alpha \in [0, 1]$. The test set contains previously unseen data samples from all encountered classes. In addition, we also showcase a baseline measure $\alpha^{\text{base}}$, highlighting the performance of each scholar in a non-continual setting, learning all classes jointly.

Furthermore, we demonstrate a forgetting measure $F_i^j$, defined for task $i$ after training $\mathcal{S}$ on $j$. This shall reflect the loss of knowledge about previous task $i$ and highlights the degradation compared to the peak performance of $\mathcal{S}$ on exactly that task:

$$F_i^j = \max_{i \in \{1,..,t-1\}} \alpha_{i,j} - \alpha_{t,j} \qquad \forall j < t. \tag{2}$$

Average forgetting is then defined as: $F_T = \frac{1}{t-1} \sum_{j=1}^{t-1} F_j^t \qquad F_t \in [0, 1]$.

### 3.3 EXPERIMENTAL SETTING

All experiments are run on a cluster of 30 machines equipped with single RTX3070Ti GPUs. Replay is investigated in a supervised CIL-scenario, assuming known task-boundaries and disjoint classes. All of the following details apply to all investigated CL algorithms, namely MIR, MerGAN, AR, ER and DGR with VAEs.

The CIL problems used for all experiments are described in Sec. 3.1. Training consists of an (initial) run on $T_1$, followed by a sequence of independent (replay) runs on $T_i, i > 1$. We perform ten randomly initialized runs for each CIL-Problem, and conduct baseline experiments for all datasets to measure the offline joint-class training performance. We set the training mini-batch size to $\beta = 100$ ($\beta = 50$ for the Fruits dataset).

For AR, selective replay of $D_i$ samples is performed before training on task $T_i, i > 1$ using the current scholar $S_{i-1}$, where $D_i$ represents the amount of training samples contained in $T_i$. For DGR, replay of $D_i$ samples is likewise performed before training on task $T_i$. This replay strategy keeps the amount of generated samples *constant w.r.t the number of tasks*, and thus comes with modest temporary storage requirements instead of growing linearly with an increasing amount of incoming tasks.

When replaying, mini-batches of $\beta$ samples are randomly drawn, in equal proportions, from the real samples from task $T_i$ and the generated/retained samples representing previous tasks. It is worth noting that classes will, in general, *not* be balanced in the merged generated/real data at $T_i$, and that it is not required to store the statistics of previously encountered class instances/labels.

## 3.4 SELECTIVE REPLAY FUNCTIONALITY

First, we demonstrate the ability of a trained GMM to query its internal representation through data samples and selectively generate artificial data that "best match" those defining the query. To illustrate this, we train a GMM layer of $K = 25$ components on MNIST classes 0, 4 and 6 for 50 epochs using the best-practice rules described in Appendix B. Then, we query the trained GMM with samples from class 9 uniquely, as described in Sec. 2. The resulting samples are all from class 4, since it is the class that is "most similar" to the query class. These results are visualized in Fig. 2. Variant generation results for deep convolutional extensions of GMMs can be found in Gepperth (2022), emphasizing that the AR approach can be scaled to more complex problems.

## 3.5 COMPARISON: AR, ER, MIR,DGR-MERGAN AND DGR-VAE

In this main experiment, we evaluate the CL performance of AR w.r.t. measures given in Sec. 3.2, and compare its performance to MIR (see Appendix E), DGR-MerGAN (see Appendix F, DGR-VAE (see Appendix C and ER (see Appendix D, since these represent principled approaches to replay-based CL. Results for AR, Er and DGR-VAE are tabulated in Tab. 1. Results for MIR and DGR-MerGAN are given in Tab. 4.

**Baseline and initial task performance** We observe superior joint training (i.e., non-CL) test accuracy $\alpha^{base}$ for DGR and ER on all datasets except Fruits where the results are identical, see Tab. 1 (bottom part). This is especially clear for experiments where the scholar is confronted with "raw" input data. A possible reasoning behind this is, that DGR and ER benefit from their internal CNN structure which is inherently capable of efficiently capturing the distribution of high-dimensional image data and becomes less prone to invariance. On the other hand, AR relies on a considerably less complex structure in its current state. Furthermore, it should be noted that DGR and ER use significantly more trainable parameters, especially when operating on raw input. For DGR, the ratio is 3.7 when applied to RGB data and 4.125 when applied to latent features. For ER, the ratio is 4.7 for RGB and 0.375 for latent features. The ability to perform well in joint-class training may also directly translate to a better starting point for CL with DGR and ER due to the initial task $T_1$ being constituted from a large body of classes in this experimental evaluation. For this reason we find the Fruits-360 dataset to be a valuable benchmark, since it is high-dimensional yet simple enough to be solved to high accuracy by AR in the baseline condition. Therefore, comparisons of CL performance are not biased by any initial difference in classification accuracy. For SVHN and CIFAR, we observe a similar situation with only minor differences to Fruits-360, as the encoded feature representations inherently have a higher degree of linear separability.

**Constant-time replay is problematic for DGR** We observe that DGR, regardless of what generator is used, performs poorly, see Tab. 1. It appears that, DGR suffers from catastrophic forgetting for all datasets under investigation. However, forgetting worsens as the number of tasks increases. This is confirmed by experiments with different task sequence lengths (D5-$1^5$, D7-$1^5$, D20-$1^5$). To a lesser extent, this is also observed for ER on e.g. FMNIST, E-MNIST. In contrast, AR is specifically

| CIL-P | D7-$1^5$A | | | | | | | | | | | | D7-$1^5$B | | | | | | | | | | | |
|---|---|---|---|---|---|---|---|---|---|---|---|---|---|---|---|---|---|---|---|---|---|---|---|---|
| measure | $\alpha_T$ | | | | | | $F_T$ | | | | | | $\alpha_T$ | | | | | | $F_T$ | | | | | |
| method | AR | ER | | DGR | | | AR | ER | | DGR | | | AR | ER | | DGR | | | AR | ER | | DGR | | |
| scenario | b. | c. | w. | b. | c. | w. | b. | c. | w. | b. | c. | w. | b. | c. | w. | b. | c. | w. | b. | c. | w. | b. | c. | w. |
| DS: MNIST | .81 | .90 | .93 | .96 | .75 | .93 | .06 | .13 | .08 | .02 | .16 | .03 | .87 | .92 | .95 | .97 | .82 | .97 | .01 | .05 | .03 | .02 | .08 | .02 |
| F-MNIST | .75 | .79 | .78 | .79 | .69 | .75 | .06 | .16 | .16 | .08 | .16 | .07 | .73 | .71 | .72 | .72 | .66 | .76 | .05 | .23 | .25 | .17 | .08 | .16 |
| FRUITS | .93 | .98 | .98 | .94 | .50 | .82 | .01 | .02 | .01 | .02 | .40 | .06 | .88 | .98 | .96 | .88 | .58 | .94 | .15 | .02 | .03 | .09 | .23 | .05 |
| SVHN | .92 | .73 | .78 | .31 | .33 | .34 | .01 | .16 | .11 | .39 | .39 | .37 | .93 | .81 | .81 | .23 | .25 | .28 | .01 | .18 | .14 | .45 | .46 | .45 |
| CIFAR-10 | .72 | .60 | .60 | .27 | .25 | .28 | .03 | .19 | .14 | .30 | .29 | .28 | .70 | .62 | .62 | .30 | .32 | .31 | .08 | .22 | .25 | .39 | .29 | .38 |
| ... | D5-$1^5$A | | | | | | | | | | | | D5-$1^5$B | | | | | | | | | | | |
| MNIST | .70 | .75 | .93 | .93 | .63 | .83 | .07 | .15 | .12 | .05 | .25 | .08 | .80 | .89 | .95 | .96 | .70 | .95 | .02 | .10 | .07 | .04 | .15 | .06 |
| F-MNIST | .70 | .75 | .75 | .78 | .63 | .68 | .16 | .31 | .27 | .28 | .71 | .27 | .71 | .71 | .70 | .61 | .56 | .63 | .17 | .36 | .38 | .37 | .42 | .33 |
| FRUITS | .93 | .99 | .99 | .92 | .40 | .48 | .11 | .03 | .05 | .09 | .22 | .20 | .82 | .99 | .98 | .92 | .44 | .81 | .13 | .01 | .02 | .07 | .21 | .19 |
| SVHN | .92 | .69 | .73 | .29 | .32 | .38 | .09 | .19 | .21 | .85 | .79 | .78 | .92 | .76 | .77 | .34 | .37 | .43 | .02 | .22 | .25 | .83 | .82 | .82 |
| CIFAR-10 | .73 | .59 | .57 | .30 | .29 | .31 | .04 | .24 | .19 | .67 | .63 | .61 | .71 | .63 | .62 | .31 | .29 | .32 | .16 | .39 | .43 | .78 | .79 | .77 |
| ... | D2-$2^5$A | | | | | | | | | | | | D2-$2^5$B | | | | | | | | | | | |
| MNIST | .83 | .88 | .88 | .94 | / | .87 | .03 | .12 | .15 | .05 | / | .09 | .73 | .96 | .96 | .97 | / | .97 | .02 | .14 | .10 | .06 | / | .08 |
| F-MNIST | .67 | .64 | .64 | .65 | / | .57 | .23 | .67 | .61 | .56 | / | .51 | .69 | .81 | .81 | .80 | / | .82 | .21 | .31 | .32 | .23 | / | .18 |
| FRUITS | .68 | .81 | .97 | .85 | / | .71 | .05 | .25 | .05 | .14 | / | .23 | .87 | .95 | .96 | .93 | / | .96 | .03 | .06 | .02 | .10 | / | .09 |
| SVHN | .92 | .64 | .65 | .20 | .23 | .25 | .01 | .25 | .27 | .88 | .91 | .88 | .92 | .89 | .89 | .63 | .61 | .62 | .04 | .29 | .32 | .91 | .93 | .92 |
| CIFAR-10 | .67 | .53 | .51 | .09 | .11 | .14 | .20 | .42 | .35 | .79 | .75 | .71 | .68 | .74 | .74 | .54 | .54 | .57 | .15 | .38 | .39 | .86 | .84 | .83 |
| ... | D20-$1^5$A | | | | | | | | | | | | D20-$1^5$B | | | | | | | | | | | |
| E-MNIST | .61 | .73 | .66 | .65 | .25 | .47 | .03 | .25 | .28 | .21 | .35 | .13 | .59 | .75 | .80 | .77 | .24 | .75 | .05 | .23 | .22 | .16 | .29 | .10 |

| DS | MNIST | F-MNIST | E-MNIST | FRUITS | SVHN | CIFAR-10 |
|---|---|---|---|---|---|---|
| method: AR | .92 | .78 | .67 | **.99** | .93 | .74 |
| DGR / ER | **.98** | **.89** | **.73** | **.99** | **.94** | **.76** |

Table 1: Main experimental results. The main table displays the results of all investigated methods (AR, DGR and ER) for each class-incremental learning problem (CIL-P) under each imposed scenario (b. = balanced, c. = constant-time, w. = weighted sample loss). We present the final test-set accuracy $\alpha_T$ and an average forgetting measure $F_T$ for each CIL-P. The relevant baselines $\alpha^{\text{base}}$ (joint-training) are showcased in the bottom table. All results are averaged across $N = 10$ runs. Detailed information about the evaluation process and experimental setup can be found in Sec. 3.2.

designed to work well when the amount of generated samples is kept constant for each task in an ever-increasing number of tasks. Figure 3 shows the development of generated sample counts over time for AR and DGR-VAE in a balanced scenario, respectively.

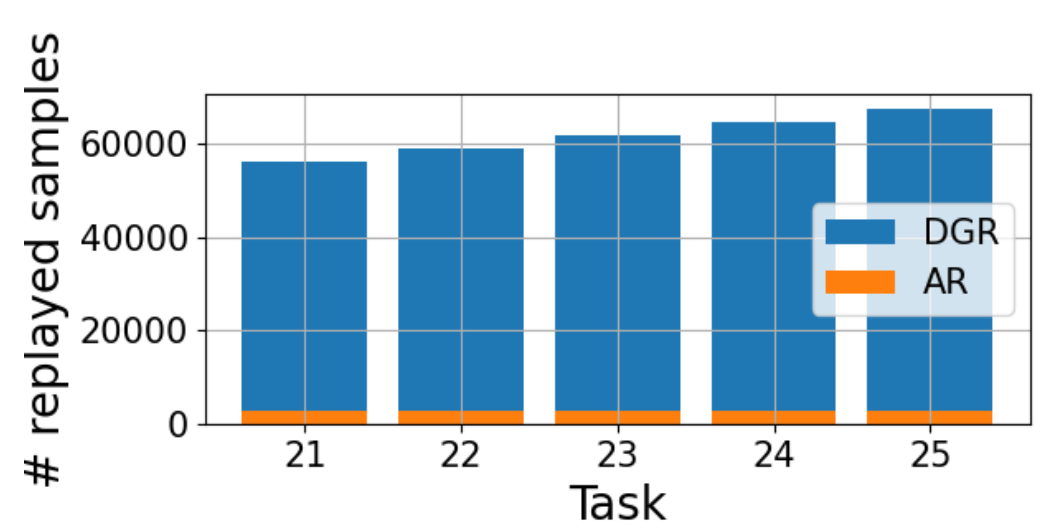

Figure 3: The amount of samples that *would* be generated per task for the E-MNIST D20-$1^5$ problem by DGR if class balancing were to be achieved, and a comparison to samples generated by AR.

**ER –vs– AR** Generally, ER shows good results on all datasets and often outperforms AR when operating on raw inputs (MNIST, FMNIST, Fruits-360 and E-MNIST datasets), although the differences are not striking, and the performance of DGR is significantly inferior still. Besides the strong differences in model complexity, a comparison between ER and AR is biased in favor of ER since AR does not get to see any real samples from past tasks. Rather, ER serves as a baseline of what can be reasonably expected from AR, and we observe that this baseline is generally quite well egalized. On the other hand, ER has the disadvantage that training time and memory usage grow slowly but

linearly with each added task, which is a unrealistic premise in practice. A fixed memory budget mitigates this problem, but has the negative effect that samples from long-ago sub-tasks will be lost over time, which will render ER ineffective if the number of tasks is large.

**AR –vs– MIR** MIr and AR share the concept of selective replay, and they both operate in a constant-time scenario although MIR has to weight generated and new samples differently in the loss. We see in general similar performance, although we must stress that MIR is highly sensitive to parameters like the weights of different terms in the loss, which must be set by cross-validation and are thus strictly speaking not compatible with CL.

**Latent replay/latent AR** For latent replay (SVHN, CIFAR), the results in the upper part of Tab. 1 show that DGR universally suffers from catastrophic forgetting although having the same baseline performance $\alpha^{\text{base}}$ as latent ER and AR. Forgetting for AR seems to only be significant for CIFAR: D5-1$^5$B after task $T_5$, due to a high overlap with classes from initial task $T_1$. Moreover, it is surprising to see that latent AR is able to achieve generally better results than latent ER. It could be argued that the budget per class for a more complex dataset like SVHN and CIFAR-10 is rather small, and it can be assumed that increasing the budget would increase CL performance. However, we reiterate that this is not trivially applicable in scenarios with a constrained memory budget.

**CF and selective replay** AR shows promising results in terms of knowledge retention and preventing CF for sequentially learned classes, as reflected by generally lower average forgetting scores. In virtually all of the experiments conducted we observed a very moderate loss of knowledge about the first task $T_1$ after full training, suggesting that AR's ability to handle small incremental additions/updates to the internal knowledge base over a sequence of tasks is an intrinsic property, due to the selective replay mechanism. Moreover, AR demonstrates its intrinsic ability to limit unnecessary overwrites of past knowledge by performing efficient *selective updates*, instead of having to replay the entire accumulated knowledge each time a task is added.

**Selective updates** As performed by AR training, are mainly characterized by matching GMM components with arriving input. Therefore, performance on previous tasks generally decreases only slightly by the adaptation of selected/similar units, as shown by the low forgetting rates for almost all CIL-P studied in Tab. 1. This implies that the GMM tends to converge towards a *trade-off* between past knowledge and new data. This effect is most notable when there is successive (replay-)training for two classes with high similarity in the input space, such as with, F-MNIST: D5-1$^5$A, where task $T_2$ (class: "sandals") and task $T_4$ (class: "sneakers") compete for internal capacity.

## 4 DISCUSSION

In summary, we can state that our AR approach clearly surpasses VAE-based DGR in the evaluated CIL-P when constraining replay to a constant-time strategy. This is remarkable because the AR scholar performs the tasks of both solver and generator, while at the same time having less parameters. The advantage of AR becomes even more pronounced when considering forgetting prevention instead of simply looking at the classification accuracy results. We may therefore conclude that AR offers a principled approach to truly long-term CL. In the following text, we will discuss salient points concerning our evaluation methodology and the conclusions we draw from the results:

**Data** Some datasets are not considered meaningful benchmarks in non-continual ML due to their simplicity. Still, many CL studies rely on these two datasets, which is why they are included for comparison purposes. SVHN and CIFAR-10 in particular are considered challenging for generative replay methods, see Aljundi et al. (2019b). E-MNIST represents a simple classification benchmark that is quite hard for CL due to the large number of classes and is well-suited to the targeted AR scenario where each task adds only a small fraction of knowledge. Finally, the Fruits-360 dataset, besides being more complex and more high-dimensional, provides a fairer comparison since it can be solved to equal accuracy by all considered methods in the baseline condition. Any differences are thus intrinsically due to CL performance.

**Pre-trained feature extractors** The use of pre-trained models is appealing in CL, since a lot of complexity can be "outsourced" to these models. As shown in Ostapenko et al. (2022), the effectiveness of feature extraction from a frozen pre-trained model relies on the relation between downstream and upstream tasks. There seems to be excellent agreement between the often-used combination of CIFAR and ImageNet, but does not extend to, e.g., the SVHN and Fruits datasets without fine-tuning.

Thus, we chose separate pre-trained models for each dataset that were optimized in a supervised fashion (SupCon) on similar but not identical data, following van de Ven et al. (2020). In contrast, self-supervised contrastive learning alleviates the need of a large labeled pre-training dataset but relies on the usage of large mini-batch sizes, as well as complex data augmentation pipelines Dwibedi et al. (2021); Chen et al. (2020a); Caron et al. (2020). We decided against such methods as they only show competitive results when combined with supervised fine-tuning on labeled data Chen et al. (2020b), or significantly increasing the total amount of classes seen in pre-training Gallardo et al. (2021).

**Time complexity of default CL methods** Regularization-based approaches like EWC have linear time complexity w.r.t. tasks, since each task adds another term to the loss function. The distillation terms in LwF ensure linear time complexity as well. Vanilla experience replay has an implementation-dependent linear time complexity since the amount of replayed samples depends on the number of previous tasks. By construction, GEM and A-GEM have linear time complexity since constraints must be computed using retained samples from all previous tasks.

**Issues with constant-time replay** Instead of achieving balance between new and recalled/generated samples by a linear increase of the latter, many recently proposed replay approaches use only a fixed number $S$ of generated or recalled samples per task. Balance is realized by a higher weight of past samples in the loss Aljundi et al. (2019b). There are several issues with this: First of all, for a large number of tasks, each task will be less and less represented in $S$ samples, making eventual forgetting inevitable, while weights for past samples grow higher and higher. Then, giving past samples a higher weight effectively increases the learning rate for these samples, which can break SGD if the weights are too high. Alternatively, the weight for the current samples can be *reduced* from its baseline value in some works van de Ven et al. (2020), ultimately leading to low learning rates and thus long training times. And lastly, the precise weights are generally set post-hoc via cross-validation Aljundi et al. (2019b); Wu et al. (2018), which is inadmissible for CL because it amounts to knowing all tasks beforehand. AR can use constant-time replay without weighting past samples due to selective updating and selective replay. We verified as well that AR, when used in a balanced scenario that linearly increases the number of samples, shows no meaningful performance differences to the constant-time case.

**Violation of AR assumptions** The assumption that new tasks only add a small contribution is not a hard requirement, just a prerequisite for sample efficiency. Based on the formalization presented in Sec. 1, its validity is trivial to verify by examining component activations of the GMM generator when faced with new data. Although we do not implement such a control strategy here, AR would simply need to replay more samples if contributions should be large. However, the chances of this happening in practice are virtually zero if the body of existing knowledge is sufficiently large.

**Initial annealing radius tuning** AR contains a few technical details that require tuning, like the initial annealing radius parameter $r_0$ when re-training with new task data. We used a single value for all experiments, but performance is sensitive to this choice, since it represents a trade-off between new data acquisition and knowledge retention. Therefore, we intend to develop an automated control strategy for this parameter to facilitate experimentation.

## 5 CONCLUSION

We firmly believe that continual learning (CL) holds the potential to spark a new machine learning revolution, since it allows, if it could be made to work in large-scale settings on real-world data, the training of models over very long times, and thus with enormous amounts of data. To achieve this important milestone, CL research must, to our mind, imperatively focus on aspects of long-term feasibility, such as also targeted in domains like life-long learning. A related aspect is energy efficiency: in order to be accepted and used in practice, training by CL must be comparable to joint training in terms of training time (and therefore energy consumption). Only in this case can the considerable advantages of CL be made to have beneficial effects in applications. In this study, we show a proof-of-concept for CL that makes a step in this direction. Namely, AR operates at a time complexity that is independent of the amount of previously acquired knowledge, which is something we also observe in humans. The overall time complexity is thus comparable to joint training. Further work will focus on the optimization of AR w.r.t. efficiency and ease of use, and the question of how to train the feature extractors in a continual fashion as well.

## 5.1 REPRODUCIBILITY

We will provide a publicly available TensorFlow 2 implementation which will be made publicly available for the camera-ready version. This repository will contain step-by-step instructions to conduct the experiments described in this article. Additional details about experimental procedures and used parameter settings are given in the various sections of the appendix (after the references) which are referenced in the text.

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

## A USE OF PRE-TRAINED FEATURE EXTRACTORS

Encoding features using pre-trained networks that transform raw-data into a higher-level and invariant representation to operate on, have shown to be beneficial for CL van de Ven et al. (2020); Hayes et al. (2020); Pellegrini et al. (2020). A current promising direction of pre-training such models is *contrastive learning*, which is performed in a supervised Khosla et al. (2020) (SupCon) or self-supervised fashion Caron et al. (2020); Chen et al. (2020a); Dwibedi et al. (2021) (SSCL). In this study, we rely on SupCon to build a robust feature extractor for more complex datasets (SVHN, CIFAR).

Here, we take a portion of the data from the target domain for pre-training, but exclude these instances from further usage in downstream CL tasks. For SVHN, we pull an amount equal to 0.5 of the total training samples from the "extra" split. For CIFAR10 we split the training set in half and use one for pre-training and the other for encoding and later usage in downstream CL. The data used to pre-train the feature extractor are thus similar but not identical to subsequent training data, following the approach of van de Ven et al. (2020).

An additional data augmentation module normalizes the input, performs random horizontal flipping and rotation in the range of $-2\% * 2\pi - +2\% * 2\pi$ for each input image. The encoder backbone is a ResNet-50 with randomly initialized weights and is trained for 256 epochs using a batch size of $\beta = 256$. No further fine-tuning is performed after pre-training. We use the normalized activations of the final pooling layer ($D = 2048$) as the representation vector.

For supervised training, a projection head is attached, consisting of two hidden layers, having a total of 2048 and 128 projection units, followed by ReLU activation. The multi-class npairs loss Sohn (2016) uses a temperature of 0.05 and is optimized via ADAM with a learning rate of $\epsilon = 0.001$, $\beta_1 = 0.9$ and $\beta_2 = 0.999$.

After pre-training we push the complete training data through the encoder network and save the output to disk for later usage. However, it would be perfectly legitimate to use the model on-the-fly to encode the data mini-batch wise, though this comes at the cost of a worse runtime efficiency.

## B AR TRAINING

AR employs a GMM scholar $L_{(G)}$ with $K = 225$ (MNIST, FMNIST, E-MNIST, Fruits) and $K = 400$ (SVHN, CIFAR) components and diagonal covariance matrices. The choice of $K$ is subject to a "the more the better" principle, and is limited only by available GPU memory.

GMM generator training follows the procedures and best-practice settings presented and justified in Gepperth & Pfülb (2021a). Training is terminated via early stopping when $L_{(G)}$ reaches a plateau of stationary loss for the current task $T_i$. We set the training epochs to 512 as an upper bound. Both, $L_{(G)}$ and the classification head are independently optimized via vanilla SGD using a fixed learning rate of $\epsilon = 0.05$. The relative strengths of component weight and covariance matrix adaptation are set to 0.1.

Annealing controls the GMM component adaptation radius for $L_{(G)}$ via parameter $r_0$. It is set to $r_0^{init} = \sqrt{0.125K}$ for the first (initial) training on $T_1$, and $r_0^{replay} = 0.1$ for subsequent (replay) tasks $T_i, i > 1$. GMM sampling parameters $S = 3$ (top-S) and $\rho = 1.0$ (normalization) are kept fixed throughout all experiments.

Other AR hyperparameters are retained to the values showcased in Gepperth & Pfülb (2021a) for all experiments, since their choice is independent of any particular CL problem at hand.

| Component | Layer | | ... | | |
|---|---|---|---|---|---|
| **Encoder** | C2D(32,5,2)-ReLU
C2D(64,5,2)-ReLU
Flatten
Dense(100)-ReLU
Dense(25)-ReLU
Dense(50) | **Decoder** | Dense(100)-ReLU
Dense((H/4)*(W/4)*64)-ReLU
Reshape((H/4),(W/4),64)-ReLU
C2DTr(32,5,2)-ReLU
C2DTr(C,5,2)-Sig. | **Solver** | Flatten
Dense(400)-ReLU
Dense(400)-ReLU
Dense(400)-ReLU
Dense(10)-Softmax |
| | | | ... | | |
| **LR-Encoder** | Flatten
Dense(1024)-ReLU
Dense(100)-ReLU
Dense(25)-ReLU
Dense(50) | **LR-Decoder** | Dense(100)-ReLU
Dense(1024)-ReLU
Dense(2048)-ReLU
Reshape(N,H,W,C) | **LR-Solver** | Flatten
Dense(1024)-ReLU
Dense(100)-ReLU
Dense(10)-Softmax |

Table 2: DNN architectures for VAE-based replay. A VAE generator consists of a mirrored encoder-decoder network. Components from the first row are utilized for MNIST, FMNIST, E-MNIST and Fruits-360. Second row components are deployed for latent replay on SVHN and CIFAR.

| ER-Solver | Layers |
|---|---|
| | C2D(32,5,1)-ReLU $\rightarrow$ MP2D(2)
C2D(64,5,1)-ReLU $\rightarrow$ MP2D(2)
Flatten $\rightarrow$ Dense(100)-ReLU $\rightarrow$ Dense(10)-Softmax |

Table 3: DNN architecture for the ER solver used for the MNIST, FashionMNIST, E-MNIST and Fruits-360 datasets. For latent replay, the LR solver network shown in Tab. 2 (bottom-right) is used.

## C  DEEP GENERATIVE REPLAY TRAINING

We implement deep generative replay (DGR) using VAEs as generators. The network structure of the generator and solver is given in Tab. 2. We choose VAEs over GANs or WGANs due to the experiments conducted in Dzemidovich & Gepperth (2022), which suggest that GANs require extensive structural tuning, which is by definition excluded in a CL scenario for all tasks but the first. Similarly, GANs and VAEs were both used in CL research, e.g., in Lesort et al. (2019) with comparable performance. The VAE latent dimension is 25, the disentangling factor $\beta = 1.0$, and conditional sampling is turned off for MNIST, F-MNIST, E-MNIST and Fruits-360 datasets, whereas it is turned on for SVHN and CIFAR to enforce that the generator naturally produces samples from previously seen classes in equal proportions. For these datasets, we also operate on latent features and use fully-connected DNNs as encoder and decoder, see Tab. 2. The learning rate for VAE generators and solvers are set to $\epsilon_G = 10^{-4}$, $\epsilon_S = 10^{-3}$ using the ADAM optimizer with $\beta_1 = 0.9$, $\beta_2 = 0.999$. Generators and solvers are trained for 100 epochs each. We reinitialize the solver network for SVHN and CIFAR before each new task, as this has shown a stabilizing effect in our empirical studies. For MNIST, FashionMNIST, E-MNIST and Fruits-360, the same structures are maintained throughout the replay training.

## D  EXPERIENCE REPLAY TRAINING

The solvers for ER are shown in Tab. 3. The ADAM optimizer is used with a learning rate of $10^{-4}$, $\beta_1 = 0.9$, $\beta_2 = 0.999$, and the network is trained for 50 epochs on each task. Analogous to the procedure for DGR, we use replay on latent feature representations, see e.g., Pellegrini et al. (2020) encoded by a pre-trained feature extractor as described in Appendix A for SVHN and CIFAR.

Similar to Riemer et al. (2018), reservoir sampling is used to select 50 samples of each encountered class to be stored. For replay, oversampling of the buffer is performed to obtain a number of samples, equal to the amount of data instances present in the current task $T_i$.

Thus, we choose an ER implementation that has constant time complexity, although the number of distinct samples per task will decrease over time. At some point, CL will break down because there are too few distinct samples per task to protect previously acquired knowledge.

# E  MIR TRAINING

We use Gen-MIR with the parameter settings for the SplitMNIST problem as described in Aljundi et al. (2019b). In order to have a fair comparison w.r.t. AR, we set the ratio of new to generated samples (n_mem) to 1, and the samples per task to 5500. For performing the experiments, we adapted the software provided by the authors to work with different MNIST splits, as well as the other datasets.

# F  DGR-MERGAN TRAINING

Our DGR-MerGAN implementation is analogous to our DGR-VAE implementation, with the exception that the generator is implemented by MerGAN instances Wu et al. (2018). For the generators, we used the network topology and the experimental settings from Wu et al. (2018). For every training mini-batch, we create a "noise" mini-batch of the same size and use it for generating samples for the discriminator and the distillation loss term. The solver is trained exactly as for DGR-VAE, see Appendix C.

# G  ADDITIONAL BASELINE RESULTS

Here, we present the experimental results for MIR and DGR-MerGAN which were hard to integrate into the main article.

Table 4: Additional experimental results for AR, MIR and DGR-MerGAN.

For each metric ($\alpha_T$ and $F_T$) and each side (A / B), methods are shown in the order AR, MIR, MerGAN with scenario columns b. / c. / w. ("/" = not applicable). (Row group label: DS.)

```
CIL-P                 D7-1^5 A                                          D7-1^5 B
measure          α_T                    F_T                     α_T                    F_T
method     AR  MIR  MerGAN      AR  MIR  MerGAN        AR  MIR  MerGAN      AR  MIR  MerGAN
scenario  b.  c.  w.  b. c. w. | b.  c.  w.  b. c. w. | b.  c.  w.  b. c. w. | b. c. w. b. c. w.
------------------------------------------------------------------------------------------------
MNIST     .81  /  .84  /  / .89 | .06  /  .05  /  / .03 | .87  /  .88  /  / .95 | .01 / .03 / / .11
F-MNIST   .75  /  85   /  / .73 | .06  /  .06  /  / .07 | .73  /  .75  /  / .73 | .05 / .08 / / .08
FRUITS    .93  /  .98  /  / .52 | .01  /  .02  /  / .30 | .88  /  .95  /  / .61 | .15 / .07 / / .21
SVHN      .92  /  .78  /  / .25 | .01  /  .03  /  / .30 | .93  /  .91  /  / .33 | .01 / .14 / / .41
CIFAR-10  .72  /  .75  /  / .33 | .03  /  .05  /  / .28 | .70 .62 .62 .30 .32 .31| .08 / .10 / / .39

...                   D5-1^5 A                                          D5-1^5 B
MNIST     .70  /  .81  /  / .85 | .07  /  .08  /  / .30 | .80  /  .95  /  / .95 | .02 / .05 / / .15
F-MNIST   .70  /  .71  /  / .69 | .16  /  .19  /  / .66 | .71  /  75   /  / .54 | .17 / .19 / / .41
FRUITS    .93 .99 .99 .92 .40 .48| .11 .03 .05 .09 .22 .20| .82 .99 .98 .92 .44 .81| .13 .01 .02 .07 .21 .19
SVHN      .92 .69 .73 .29 .32 .38| .09 .19 .21 .85 .79 .78| .92 .76 .77 .34 .37 .43| .02 .22 .25 .83 .82 .82
CIFAR-10  .73 .59 .57 .30 .29 .31| .04 .24 .19 .67 .63 .61| .71 .63 .62 .31 .29 .32| .16 .39 .43 .78 .79 .77

...                   D2-2^5 A                                          D2-2^5 B
MNIST     .83 .88 .88 .94 .00 .87| .03 .12 .15 .05 .00 .09| .73 .96 .96 .97 .00 .97| .02 .14 .10 .06 .00 .08
F-MNIST   .67 .64 .64 .65 .00 .57| .23 .67 .61 .56 .00 .51| .69 .81 .81 .80 .00 .82| .21 .31 .32 .23 .00 .18
FRUITS    .68 .81 .97 .85 .00 .71| .05 .25 .05 .14 .00 .23| .87 .95 .96 .93 .00 .96| .03 .06 .02 .10 .00 .09
SVHN      .92 .64 .65 .20 .23 .25| .01 .25 .27 .88 .91 .88| .92 .89 .89 .63 .61 .62| .04 .29 .32 .91 .93 .92
CIFAR-10  .67 .53 .51 .09 .11 .14| .20 .42 .35 .79 .75 .71| .68 .74 .74 .54 .54 .57| .15 .38 .39 .86 .84 .83

...                   D20-1^5 A                                         D20-1^5 B
E-MNIST   .61 .73 .66 .65 .25 .00| .03 .25 .28 .21 .35 .00| .59 .75 .80 .77 .24 .00| .05 .23 .22 .16 .29 .00
```

