# OpenReview forum: "Adiabatic replay for continual learning"
_ICLR.cc/2024/Conference — Submitted to ICLR 2024_

### Official Review · Reviewer_po4B · 2023-10-28

**Soundness:** 2 fair
**Presentation:** 3 good
**Contribution:** 2 fair
**Rating:** 3
**Confidence:** 5

**Summary:**

The paper studies the problem of continual learning. The paper focuses on the generative replay approach for continual learning and proposes a new setting where the new information from the incoming task is not too significant but rather incremental based on the previous tasks (the AR setting). The paper proposes to use a Gaussian mixture model to simultaneously act as the generator as well as the learner for continual learning under the AR setting. The paper shows superior performance of the proposed method under some of the restricted settings.

**Strengths:**

1. The paper proposes a new setting that might be interesting to inspire new research direction where the information of the new incoming task is not too significant compared to the old ones.

2. The paper uses a Gaussian mixture model, which can be used both as the generator and the learning model, and shows improved performance on multiple settings.

**Weaknesses:**

1. The paper lacks justification for the adiabatic assumption. The paper creates specific settings for the experiments and makes some discussions, but are there many real-world scenarios in which such adiabatic assumptions can be applied? Also, there seems to require a more detailed formulation of such an adiabatic assumption. Say if the new information is too little, then it gets back to the ordinary training and there is no reason to do continual learning. Some mathematical formulation of the adiabatic assumption should be defined.

2. In addition to the mathematical definition of the adiabatic assumption, it is also not clear why the selected settings in the experiments satisfy such an assumption. Why the tasks are chosen in such a way? From a more practical perspective, if we are dealing with real-world tasks, how do we check if the incoming task satisfy the adiabatic assumption and we can use the proposed method?

3. The paper only tests for certain restricted settings and uses a relatively simple model (GMM). Though the paper mentions that there could be more advanced version of the GMM model that could solve the capacity problem, but there does not seem to be much evidence to support such a claim.

4. It is claimed that the proposed method does not have the problem of scaling up as the number of tasks increases. However, for GMM, there is the number of clusters and I wonder should such a number be set according to the total number of tasks? If there are infinite number of classes coming in a stream, should the number of clusters also increase? Even though the change of size could be small, if we want to use a much more capable model as mentioned in the paper, will the model size go up as the tasks increase?


Minor:
Page 1: "On the one hand, there are ”true” replay" (the quote symbols)

**Questions:**

Please check the weaknesses part.

After rebuttal update:
Thanks to the authors for the detailed response. My concerns about the paper remain, mainly about the precise definition of the adiabatic assumption. I keep my score unchanged.

---

> ### Author Response · Authors · 2023-11-12
> **Rebuttal reply**
>
> Concerning your comments:
>
> 1a. In virtually all articles about CL, the same amount of samples is added for each task, so the adiabatic assumption is automatically fulfilled after a few tasks. E.g., a 10-fold split of CIFAR along class boundaries will add 3000 samples per task, so at task t>=2 the number of new samples is smaller than the number of previous samples. So we feel the adiabatic assumption is adhered to to good approximation.
> Of course you could imagine tasks where this assumption is always violated, but we argue that learning in humans or robots get more and more adiabatic over time, since additions tend to have the same size and the pool of existing knowledge is ever-growing.
>
> 1b. We will add a more formal statement of the adiabatic assumption. We will also add experiments on 5-fold and 10-fold equal splits to show that a violation does not break AR.
>
> 2. The tasks are chosen such that the a.a. is always fulfilled, and are otherwise modeled after common continual learning benchmarks. As stated in 1b, a violation of the a.a. does not break AR, and we will show this by adding more experiments. From a practical perspective, it is very easy to detect whether a new task violates the a.a.: we just need to measure how many different existing GMM components are best-matching ones when exposed to the new task's data. If there are few of them, then the a.a. is valid. This will be discussed as well.
>
> 3. We will remove the section about deep GMMs as this is not followed up in the article. Yes, there are deep convolutional extensions to GMMs that can be used, but thanks to latent replay, the flat GMMs do a sufficiently good job as shown by the experimental results.
>
> 4a. No, our GMM does not grow over time. Instead, we choose K as high as our memory allows it because, as stated in app. B, more is always better when it comes to choosing K. This was shown in Gepperth&Pfülb(2021). If the number of classes is higher than K, performance will suffer, to be sure. So yes, we need to know or guess the total number of classes in advance, but any other DNN model needs to do this as well.
>
> 4b. Even the most sophisticated DNN will fail for an infinite number of classes. Virtually *all* recent works on CL that we are aware of assume that the total number of classes is known beforehand, and all works use static DNNs/CNNs that do not grow in size. A priori knowledge of the number of classes is required for tuning generator and solver structure, as well as for choosing the size of the classification head. So our method is no different in that respect.

---

> ### Author Response · Authors · 2023-11-23
> **Paper changes**
>
> Dear reviewer,
> this is what we adapted in the revised version of the paper. We hope this reflects the issues you raised, in any case it has made the paper a whole lot more solid.
>
> - formalized the assumptions AR makes in the introduction, and how to measure whether they are fulfilled. We also made it clear that AR does not "break" if these assumptions are not fulfilled.
>
> - included MIR and DGR-MerGAN as baselines. More was not possible to achieve given the time constraints.
>
> - introduced re-weighting of prior data in the loss for VAE-DGR and ER
>
> - performed where AR is used in a balanced replay scenario, i.e., the nr of samples increases linearly with the task as in vanilla deep generative replay. Since this did not really change the results in any way, we did not include these experiments in the results table. Instead, we added a paragraph to the discussion.
>
> - adapted our use of the term "foundation models" in favor of pre-trained feature extractors
>
> - discussed differences notably to MIR in the discussion section
>
> - compared constant-time replay in AR to how it is generally performed in related work
>
> - discussed additional related work, mainly  Klasson et al. (2023, TMLR; https://openreview.net/pdf?id=Q4aAITDgdP) and McClelland et al. (2020, Phil Trans R Soc B; https://royalsocietypublishing.org/doi/full/10.1098/rstb.2019.0637) as conceptual foundations of our approach
>
> - added pseudocode for AR
>
> Thanks again for contributiong your remarks!

---

### Official Review · Reviewer_m3N7 · 2023-10-30

**Soundness:** 2 fair
**Presentation:** 2 fair
**Contribution:** 3 good
**Rating:** 5
**Confidence:** 4

**Summary:**

This paper first describes an important issue with replay in continual learning: whenever learning something new, it is generally required to replay all past tasks to avoid forgetting. This means there is an unbounded linear growth of to-be-replayed samples. The paper then proposes that this might be addressable with *adiabatic replay*: when learning something new, we should only replay samples – and only update the parts of the network – that are closely related to the newly learned information. To provide a proof-of-principle demonstration of this idea, the authors turn to GMMs in which they only replay and update the mixture component(s) most similar to the new data samples.

**Strengths:**

The paper addresses an important problem in an original way. It is a fundamental limitation of current replay approaches that all tasks thus far need to be replayed. In the last few years there have been several continual learning studies that explored whether benefits could be gained by deciding what to replay in a smart way, but generally the conclusion of these studies has been that it is very hard to do better than simply doing balanced, random sampling from all tasks so far. This study takes a novel and promising approach to this problem by using GMMs, in an attempt to provide a proof-of-principle demonstration that it is possible to do better.

**Weaknesses:**

**Issue 1: important comparisons are missing**

I am afraid critical comparisons are missing for a convincing proof-of-principle demonstration that selective replay can provide substantial benefits. The paper compares AR (using GMMs) with DGR and ER (both using VAEs), and because AR performs better than DGR, the authors conclude that selective replay is beneficial. But I do not think that this can be concluded from this comparison. There are several important differences between AR and DGR – not only the use of selective replay, that could explain the difference in performance. Perhaps the authors would argue that DGR is a “state-of-the-art” technique, and that showing improved performance with a method using selective replay would be enough demonstration, but I do not agree with that. Firstly, it is not clear how DGR is implemented and it seems this is not done in an optimal way (see issues 2 and 3). Secondly, the performance obtained by DGR is simply not good (e.g., for MNIST it is substantially lower than a linear classifier could obtain). Improvements on top of that are thus not necessarily meaningful.
I would like to encourage the authors to instead include a comparison that directly assesses the benefit of using selective replay over normal replay (e.g., AR against the exact same version but using normal replay).

**Issue 2: relevant past work is not discussed (appropriately)**

Discussion of past literature is insufficient, and some important aspects are ignored. Firstly, it is claimed at several places that current replay methods must replay an amount of samples proportional to the number of past tasks, but Van de Ven et al. (2020) empirically showed that it is possible to do better than this because “preventing forgetting is easier than learning”. Although to make this possible, it is important that the loss terms from the replayed data and the current data are balanced appropriately. (It is also unclear to me whether this is done in the current study when a limited amount of data is replayed with DGR and ER, see also the last question under issue 3 below.) Another paper that would be good to discuss is Klasson et al. (2023, TMLR; https://openreview.net/pdf?id=Q4aAITDgdP), as they show that benefits can be obtained by being smart about what to replay. I think it is also relevant to discuss McClelland et al. (2020, Phil Trans R Soc B; https://royalsocietypublishing.org/doi/full/10.1098/rstb.2019.0637), as they make a conceptual argument that it should not be necessary to replay everything.

While I think it is important that these past studies on this topic are appropriately discussed, I do not think that the existence of these studies means that the novelty of the current study is compromised, as the current study approaches the problem in a novel and original way. In fact, these past studies illustrate the importance of the topic that is addressed by the current study.

**Issue 3: insufficient experimental details**

- How is the solver of AR trained? Is the solver also trained with “adiabatic replay”? Or is the solver trained with “regular replay”, and suffers from a linear increase in the amount of replayed data? How are the labels for the replayed data obtained?

- For DGR, how are the labels obtained for training the solver?

- For ER and DGR, how are the loss terms from the replayed data and the current data weighed? Are they balanced as in Van de Ven et al. (2020), or are they simply added?

Minor issues:

- Many of the in-text citations are formatted incorrectly

- Towards top of p3: Maximally interference replay -> Maximally interfered replay

- Bottom of p4: the notation DX-Y is used, but has not (yet) been introduced

- Given the importance of ER in the current manuscript, the buffer size that is used should be mentioned in the main text

**Questions:**

For the main issues to address, please see the first three issues raised under “Weaknesses”.

As I think this paper takes an original angle to an important problem, I hope the authors will be able to address these raised issues.

I would be happy to actively engage in the discussion period.

---

> ### Author Response · Authors · 2023-11-12
> **Rebuttal reply**
>
> Dear reviewer, thanks for the constructive review! Here are some of our thought w.r.t. the weaknesses you raised.
>
> 1a. All DGR and ER experiments are conducted in the constant-time setting, i.e., as many generated/replayed samples as there are new samples.  Normally, one would train DGR in a balanced fashion, i.e., generate more samples for each new task. Since we do this differently, the results for DGR are "not good", as you remarked. However, this ensures that the comparison to AR is fair, and this is why we conclude that selective  replay is beneficial since DGR obviously cannot cope with constant-time replay.
>
> 1b. To show that our DGR implementation is sound, we will include results when  DGR is trained in the  standard fashion. We actually did this long ago, and results are indeed stronger and more in agreement with the literature.
>
> 1c. We will compare  AR to the exact same version but with normal replay. This will probably not change the results, and thus prove that selective replay performs similarly to normal replay, but at far better efficiency.
>
> 2. We will discuss these references appropriately.
>
> 3. We will include these experimental  details which are indeed a bit shakily described.
> * The AR "solver", which is just a linear classifier, is trained with AR as well and thus does not compromise AR's sample efficiency.
> * The labels are obtained by running the generated samples through the solver. We experimented with conditional replay but found that this enormously impaired the quality of generated samples.
> * The ER buffer  size is 50.
> * The loss terms are not weighted: all generated samples have the same weight in the loss. The reason: we found at several occasions that specific weights for each replay task can lead to complications. As more and more tasks are added, the weights for long-ago tasks must be chosen higher and higher. If coefficients are too high for a given task, gradient descent can fail completely because the effective learning rate is too high. In addition, we feel it would be unfair if task-specific weights were used for DGR and ER, since they are neither used nor required for AR, resulting in a biased comparison. We can of course include such results for reference.

---

> > ### Comment · Reviewer_m3N7 · 2023-11-15
> > **Response to initial author rebuttal**
> >
> > Thank you for the fast response. At the moment, I have two questions/comments in response.
> >
> > {1} Could you provide some intuition why it is OK for the AR "solver" to be trained with AR? Why does this solver not suffer from the problem of data imbalance?
> >
> > {2} I do not agree with your claim that it would be "unfair" to use "task-specific weights" for DGR and ER. This approach does not use additional information, it is simply a way to decouple the amount of samples that is replayed and the relative importance that they are given. I agree that using such task-specific weights can have problems as well -- that is one of the reasons why I am excited about the approach proposed in your paper, but it seems clear that using such task-specific weights is a more sensible approach than letting the weight of the replay loss being determined by the amount of samples that can be replayed. So I think that using the task-specific weights is the more important baseline relative to which to demonstrate improvements.
> >
> > Otherwise, I look forward to seeing the outcome of points 1c and 2.

---

> > > ### Author Response · Authors · 2023-11-16
> > > **Reply to response :-)**
> > >
> > > Dear reviewer,
> > >
> > > concerning question (1): GMMs have a sparsity property. This means that, to very high precision, there is nearly always exactly one component which has responsibility/posterior probability close to 1, and the others components' responsibilities are very close to zero due to normalization. In such a setting, when using a linear classifier without bias (as we do), a weight $W_{ij}$ from component i to class output j will not be adapted if component i has a posterior close to 0. Therefore, weights to components whose posterior is always 0 (because the samples that would activate them are not replayed) will not be affected by imbalances.
> > >
> > > Concerning (2): we will measure the baselines for ER and DGR with task-specific weights as proposed in Van de Ven et al. (2020), it really does make sense to use baselines that are widely known and accepted. We will provide a discussion of the pros and cons of such an approach and justify why we choose not to do this for AR.

---

> ### Author Response · Authors · 2023-11-23
> **Paper changes**
>
> Dear reviewer,
> this is what we adapted in the revised version of the paper. We hope this reflects the issues you raised, in it has made the paper a whole lot more solid. And btw: thanks a lot for those two references, namely the McClelland&al., paper we did not have that on our radar...
>
> - formalized the assumptions AR makes in the introduction, and how to measure whether they are fulfilled. We also made it clear that AR does not "break" if these assumptions are not fulfilled.
>
> - included MIR and DGR-MerGAN as baselines. More was not possible to achieve given the time constraints.
>
> - introduced re-weighting of prior data in the loss for VAE-DGR and ER
>
> - performed where AR is used in a balanced replay scenario, i.e., the nr of samples increases linearly with the task as in vanilla deep generative replay. Since this did not really change the results in any way, we did not include these experiments in the results table. Instead, we added a paragraph to the discussion.
>
> - adapted our use of the term "foundation models" in favor of pre-trained feature extractors
>
> - discussed differences notably to MIR in the discussion section
>
> - compared constant-time replay in AR to how it is generally performed in related work
>
> - discussed additional related work, mainly  Klasson et al. (2023, TMLR; https://openreview.net/pdf?id=Q4aAITDgdP) and McClelland et al. (2020, Phil Trans R Soc B; https://royalsocietypublishing.org/doi/full/10.1098/rstb.2019.0637) as conceptual foundations of our approach
>
> - added pseudocode for AR
>
> Thanks again for contributing your remarks!

---

### Official Review · Reviewer_ew63 · 2023-10-31

**Soundness:** 2 fair
**Presentation:** 1 poor
**Contribution:** 2 fair
**Rating:** 3
**Confidence:** 5

**Summary:**

This paper focuses on replay-based approaches to continual learning and proposes a new method called AR which results in the following contributions:

1. Selective replay: Previous knowledge is not replayed indiscriminately, but only where significant overlap with new data exists.

2. Selective update: Previous knowledge is only modified by new data where an overlap exists.

3. Near-Constant time complexity.

**Strengths:**

The high time cost of replay methods is a significant problem in continual learning. I appreciate the motivation of this paper which tries to solve this problem by GMM.

**Weaknesses:**

1. The writing needs to be improved.

2. I usually don't focus too much on the experiments, but i do believe that the experiments should be improved where more baselines and time cost experiments are necessary.

3. The reasonable way of chosing hyperparameter $K$ is necessary. Please discuss more about $K$.

4. Lack of novelty. The key technique in this paper is well-known.

5. It seems that the main text of this paper is over the limit of 9 pages, where Section 5.1 is in page 10.

Details of my concerns are listed in Questions.

**Questions:**

First, I am concerned about the writing and the presentation of this paper. Specifically, the descriptions of proposed method should be transparency and easy to follow. It is better to show your method by pseudo codes.

Second, I am concerned about the proposed method AR.

1. Is the hyperparameter $K$ fixed and how to determine a reasonable $K$?

2. In my view, one component of GMM corresponds to one distribution of several classes. When $K>$ the number of seen classes, is it possible to chose an unknown component at the query step and what dose the generated samples look like in this scenario? It seems that the ability of preventing forgetting is determined by $K$. When there are so many classes whose number $\gg K$, it is hard to prevent forgetting due to the overlapping. If I was wrong, please correct it.

 Thrid, I usually don't focus too much on the experiments, but i do believe that the experiments should be improved.

1. There are only two old replay methods (DGR and ER). It is better to compare it with more and new replay baselines.

2. In my opinion, the potential low time cost of proposed method is a significant advantage. It is better to demonstrate this by more time cost experiments.

3. The splited tasks $T_2, T_3, \dots$ contain only one class. How dose the proposed method perform if there are more classes in one task?

---

> ### Author Response · Authors · 2023-11-12
> **Rebuttal reply**
>
> Dear reviewer,
> thank you for your evaluation of our paper, this is valuable  advice for improving the paper.
> In order to do this, we would ask you to provide a bit more information w.r.t. a few points you raise.
>
> Weaknesses:
> 1. we will add pseudocodes, no problem. Could you be more specific in what ways the writing should be improved?
> 2. What baselines are you referring to? Could you provide references?
> 3. As stated in appendix B, we use the best-practice procedures from Gepperth&Pfülb(2022) for choosing the hyperparameters of the GMM. In this reference, it is shown through experimentation that K follows a "the more the better" principle. This is also stated in appendix B, but we will include this info and a citation in the main text.
> 4. You do not seem to provide a reference for your claim, what works are you referring to?
> 5. It is clearly stated in the ICLR 2024 author's guidelines that the reproducibility statement does not count towards the page limit. Please be sure to take this into account in your evaluation.
> ---
> Questions concerning AR:
> 1. see above
> 2. It is true that K must be >= #(classes), so there is at least one component per class. So we need to guess the total #(classes) beforehand. But: with DNNs, we need to choose appropriate layer sizes beforehand, too, so the situation is the same. For solvers  and generators, this is usually done by cross-validation. For our method, no cross-validation is required.
>
> Questions concerning experiments:
> 1. again, we would be grateful for references here. There are many variants of generative replay, some are very specific to certain settings.
> 2. We will include a table with runtime measurements. The time cost will be basically proportional to the nr. of samples in Fig. 4.
> 3. There is no problem at all with more than one class per task. We will include splitMNIST and splitCIFAR (5-fold split with two classes per split) experiments to demonstrate this.

---

> > ### Comment · Reviewer_ew63 · 2023-11-12
> > **More comments**
> >
> > Thank you for your reply. According to your responses, I provide some additional comments.
> >
> > 1. To my knowledge, I recommend some replay baselines: MeRGAN [1], SPB [2], PASS [3]. I think more comparisons will reflect more advantages of your proposal.
> >
> > 2. The key technique is GMM which is well-known. It is likely that this paper is a A+B work which leverages GMM in CL. It is OK but not surprising.
> >
> > 3. I am still a bit concerned about the proposed AR.
> >
> > a. Since $K$ must be $\geq$ #(classes), is it possible to chose an unknown component at the query step and what dose the generated samples look like in this scenario?
> >
> > b. It seems that the number of classes are fixed before the entire training. This may indicate that you implement your method by fixing the class number. I think this is not a valid way of implementing the class incremental scenario. Please refer to the implementing way of [4].
> >
> > [1] Memory Replay GANs: learning to generate images from new categories without forgetting. NeurIPS 2018.
> >
> > [2] Striking a balance between stability and plasticity for class-incremental learning. ICCV 2021.
> >
> > [3] Prototype augmentation and self-supervision for incremental learning. CVPR 2021.
> >
> > [4] Mnemonics Training: Multi-Class Incremental Learning without Forgetting, CVPR 2020.
> >
> > If authors considered my comments and improved this work well, I think this paper is worth of rating 6.

---

> ### Author Response · Authors · 2023-11-16
> **Answers to reviewer questions**
>
> Dear reviewer,
> thanks a lot for your clarification, this is very helpful. Concerning your remarks:
>
> (1a) we will discuss references [1],[2],[3].
>
> (1b) we do not seem to be able to get the code in the repositories provided in [1,2,3] to run (for SPB, the provided repository seems to be for a different paper), so we are currently re-implementing MerGAN and SPB. To compensate for this, we include baseline measurements for Maximally Interfered Retrieval (NeurIPS 2019).
>
> (3a) At every task, all components are used, so it is not possible to choose an unknown/unconverged component. For each subsequent task, components will be partially re-allocated to describe new classes.
>
> (3b) We do not preallocate components to certain classes (this is how we interpret your question,did we understand you correctly?). Neither do we make any other assumptions regarding classes other than the maximal number of classes that will occur. This is a common assumption that all other works make, even [4]. Essentially, this information is needed to determine the size of the single-head classifier (which we use as well, not a multi-head one).
> Component re-allocation for each new task is a dynamic process where AR adapts those components which are most similar to the new tasks' data. And leaves the ones that are dissimilar unchanged. However, components are not simply overwritten but must now interpolate between past and new knowledge.

---

> ### Author Response · Authors · 2023-11-23
> **Paper changes**
>
> Dear reviewer,
> this is what we adapted in the revised verison of the paper. We hope this reflects the issues you raised, in any case it has made the paper a whole lot more solid.
>
> - formalized the assumptions AR makes in the introduction, and how to measure whether they are fulfilled. We also made it clear that AR does not "break" if these assumptions are not fulfilled.
>
> - included MIR and DGR-MerGAN as baselines. More was not possible to achieve given the time constraints.
>
> - introduced re-weighting of prior data in the loss for VAE-DGR and ER
>
> - performed where AR is used in a balanced replay scenario, i.e., the nr of samples increases linearly with the task as in vanilla deep generative replay. Since this did not really change the results in any way, we did not include these experiments in the results table. Instead, we added a paragraph to the discussion.
>
> - adapted our use of the term "foundation models" in favor of pre-trained feature extractors
>
> - discussed differences notably to MIR in the discussion section
>
> - compared constant-time replay in AR to how it is generally performed in related work
>
> - discussed additional related work, mainly  Klasson et al. (2023, TMLR; https://openreview.net/pdf?id=Q4aAITDgdP) and McClelland et al. (2020, Phil Trans R Soc B; https://royalsocietypublishing.org/doi/full/10.1098/rstb.2019.0637) as conceptual foundations of our approach
>
> - added pseudocode for AR
>
> Thanks again for contributing your remarks!

---

### Official Review · Reviewer_M6q9 · 2023-11-02

**Soundness:** 2 fair
**Presentation:** 2 fair
**Contribution:** 2 fair
**Rating:** 3
**Confidence:** 4

**Summary:**

The paper proposes adiabatic replay, a method for replay in continual learning that tailors the replay sample retrieval step to the incoming samples to be learned. This approach can theoretically avoid replaying all previous tasks, by focusing only on the modes of the distribution which overlap the most with the new task in the model's latent space. The proposed approach is a variant of deep generative replay, using a VAE with a GMM prior as a generative model. The authors argue that this choice of prior is better suited for *variant generation* which enables sampling of datapoints most likely to be interfered from the new task. The authors explore their method in offline class incremental settings, comparing to ER and DGR.

**Strengths:**

1.  The motivation for the paper is sound : there is significant value in carefully selecting the replay data and avoid full replay of past samples, as the latter is very computationally demanding.

**Weaknesses:**

1.  Overall, the paper's empirical execution is weak. The experimental protocol is limited to small sequences of tasks using arbitrarily small models. There is no discussion about the computational cost of the method. This is disappointing, as I do believe that the essence of the method has potential in different settings (for example targeting compute restricted settings, with pretrained models).
2. The baseline numbers are very weak, which raises concerns about the empirical rigor of the work. For example, deep generative replay with a VAE of similar complexity in MIR [1] gets 80% on split mnist, *trained online* with 2 classes per task, which is arguably a much more difficult setting than the one in the paper. On that note, the paper would benefit from including this baseline in the paper.
3. To the best of my understanding, the conceptual differences with MIR are small : the idea of fetching points close in the VAE's latent space is exactly what MIR does. In other words, AR is similar to Gen-MIR where 0 gradient ascent steps to maximize interference are done.
4. The paper could benefit from some tweaks in the main paper. For example, it is unclear at which level latent replay operates; adding a detailed figure would greatly aid understanding. Moreover, I think the authors' interpretation of Foundation Models are merely any pretrained model, rather than a "generalist" model trained on vast, diverse data.

**Questions:**

1. What is the buffer size used for ER ?
2. When is latent replay (vs standard replay) performed ? it seems that only SVHN and CIFAR use latent replay ?

---

> ### Author Response · Authors · 2023-11-12
> **Rebuttal reply**
>
> Dear reviewer,
> there are some points to be made w.r.t. weaknesses you address:
>
> 1a. We are not sure what you mean by "arbitrarily small models". The baseline models, e.g., in VAE-replay or the solvers in ER, are comparable to those used in related work. The GMMs are complex enough to solve the task at hand, which is all that should matter.
>
> 1b. The "hardness" of a CL task is mainly due to the number of tasks, not due to their inherent complexity. By selecting CL tasks that at least 5  additional tasks after the first one, we ensure a sufficient level of difficulty. The usual tasks in the literature, like 2-2-2-2-2 splits on CIFAR or MNIST, can be included without any problems. We will update the paper accordingly.
>
> 1c. Fig. 4 directly addresses the computational cost. We do not measure execution times because those are very dependent on hardware, but complexity as a function of the # of tasks. This will be directly proportional to actual execution times.
>
> 2. The baselines for VAE-replay are not comparable to those reported elsewhere, since we are training in the compute-restricted constant-time setting. I.e., we generate a constant number of samples for each task, instead of increasing this number at each task, see also Fig. 4. If we used a balanced replay strategy, of course the numbers would be a bit better but it would no longer be a fair comparison.
>
> 3. The key difference to MIR is selective updating, which the VAEs in GEN-MIR cannot do. They have to be trained with a balanced dataset. The idea of selective replay is of course quite similar, although our method does not require gradient descent to do it.
>
> 4. Point taken, this will be incorporated.
> -----
>
> Concerning your questions:
> 1. Buffer size is 50
> 2. Correct, latent replay is done for CIFAR and SVHN.

---

> > ### Comment · Reviewer_M6q9 · 2023-11-16
> > **Discussion**
> >
> > Thank you for your answers.
> >
> > 1a. I respectfully disagree; I don't think that getting 80% accuracy on MNIST tasks is solved.
> > 1c. (please correct me if I am wrong) Fig. 4 indicates how many samples are required to create a balanced stream. What I would like to see instead is how many samples are required for DGR and AR to reach the same performance *in practice*.
> >
> > 2. In most existing online CL literature (e.g. MIR), approaches do operate in the compute-restricted constant-time setting, where the number of replay steps or replay compute does not grow as a function of number of tasks seen. A standard practice is to allocate 50% of the compute to replay and 50% to learning the current task. Hence I don't see why the numbers in your setting (which does multiple epochs and is more compute hungry) shows lower numbers.
> >
> > 3. Again, this is *false*. MIR does not build balanced datasets.
> >
> > Thank you for clarifying my other questions.

---

> ### Author Response · Authors · 2023-11-19
> **Discussion**
>
> Thank you for your comments, we are glad to start a discussion.
> Concerning the issues you raise:
>
> 1a. Well, the task is not solved, obviously, but 80% on MNIST in the compute-restricted setting is what many other approaches obtain, too. MIR, for example, does 82% on average on MNIST for the 2-2-2-2-2 split. So we believe our models in general do not do worse than those of other approaches. Hence we believe we do not use "arbitrarily small models" but models of similar capacity as other approaches do.
>
> 1c. Point taken, we could determine the amount of samples to be replayed per task via cross-validation. We do not do this because cross-validation in CL is not really admissible (even if many articles use it) since it requires advance knowledge of all tasks. If we do not make use of such a "look into the future", one can reasonably use either the constant-time setting or the balanced setting.
>
> 2.Our issue with the constant-time settings used in other approaches is that samples are replayed indiscriminately, even though the number of samples remains constant. So, over time, the number of replayed samples for each past task will go to 0, at which point forgetting is ensured.
>
> 3. Thank you for pointing this out! Indeed MIR uses a constant-time replay strategy, we will adapt this part of the text. The balancing in MIR is achieved by giving replayed samples a higher weight in the loss. However, the weight (3.0 in the case of MNIST 5-fold split) has to be chosen via cross-validation over all tasks, which we consider problematic in CL (see above). it is highly sensible to the task setup: if we train MIR on MNIST using a 10-fold split with the same settings, the final accuracy drops to 65%, with the "oldest" classes showing very high forgetting. So this balancing-via-loss-weights approach has issues as well, in addition to issues mentioned in 2. In our approach, the generated and the current samples have the same weight in the loss, which avoids tuning this  hyper-parameter.

---

> ### Author Response · Authors · 2023-11-23
> **Paper changes**
>
> Dear reviewer,
> this is what we adapted in the revised verison of the paper. We hope this reflects the issues you raised, in any case it has made the paper a whole lot more solid.
>
> - formalized the assumptions AR makes in the introduction, and how to measure whether they are fulfilled. We also made it clear that AR does not "break" if these assumptions are not fulfilled.
>
> - included MIR and DGR-MerGAN as baselines. More was not possible to achieve given the time constraints.
>
> - introduced re-weighting of prior data in the loss for VAE-DGR and ER
>
> - performed where AR is used in a balanced replay scenario, i.e., the nr of samples increases linearly with the task as in vanilla deep generative replay. Since this did not really change the results in any way, we did not include these experiments in the results table. Instead, we added a paragraph to the discussion.
>
> - adapted our use of the term "foundation models" in favor of pre-trained feature extractors
>
> - discussed differences notably to MIR in the discussion section
>
> - compared constant-time replay in AR to how it is generally performed in related work
>
> - discussed additional related work, mainly  Klasson et al. (2023, TMLR; https://openreview.net/pdf?id=Q4aAITDgdP) and McClelland et al. (2020, Phil Trans R Soc B; https://royalsocietypublishing.org/doi/full/10.1098/rstb.2019.0637) as conceptual foundations of our approach
>
> - added pseudocode for AR
>
> Thanks again for contributing your remarks!

---

### Meta-Review · Area_Chair_59aw · 2023-12-06

**Metareview:**

The paper proposes a (generative) replay strategy, referred to as adiabatic replay, which is based on selective replay to alleviate catastrophic forgetting in continual learning under compute and memory constraints. Conceptually, the paper addresses an important problem of replay methods being potentially expensive and needlessly wasteful in their selection and replay strategy. However, the practical presentation of the paper has resulted in several concerns being raised by all reviews, which has sparked discussion among all reviewers.

Primarily, the concerns revolve around the novelty of the given method, its relation to several existing very related strategies (such as MIR), and the limited experimental comparison both in terms of exhaustiveness and rigor. In the discussion the authors have addressed a subset of these points, attributed to the limited amount of time. For instance, more baselines have been added (in the appendix) and clarification texts have been included in the main paper.

These revisions do in fact improve the paper. However, the AC agrees with the reviewers that much more is required for the paper to be published at a conference such as ICLR. In particular, the reviewers have raised concerns about the fairness of comparison and the way related techniques are employed/described. From the discussion and revisions, it did not become apparent why some scenarios are deemed appropriate for AR, but not for other methods (e.g. task-specific weighting). This has in turn also sparked discussion on the novelty of the method w.r.t  for instance MIR, as the empirical analysis also does not sufficiently support claims surrounding e.g. compute efficiency. In the present form it is thus not fully clear whether the contribution of the paper lies in nuances that are insufficiently graspable  and how much overlap there is with existing prior works that require more thorough discussion in the paper. To quote one of the reviewers "The key technique is GMM which is well-known. It is likely that this paper is a A+B work which leverages GMM in CL. It is OK but not surprising". The AC agrees that there exists value in taking well-known approaches and bringing them into new fields, but in the present paper it remains unclear why this makes for an entirely new method and contribution, given that the experimental one is limited.

 The AC thus recommends a major revision and resubmission of the paper.

**Justification For Why Not Higher Score:**

The feedback given by the reviewers was thoughtful, detailed and constructive in nature. While some of the points that were raised could already be addressed by the authors, the limited discussion and revision time at ICLR was simply insufficient to make all the required changes to the paper prior to publication. The AC thanks both the reviewers and authors for fruitful discussion and hopes that the suggested feedback can be accommodated in a resubmission to a future venue.

**Justification For Why Not Lower Score:**

N/A

---

### Decision · Program_Chairs · 2024-01-16

Reject